# Acute Neuromuscular Responses to Whole-Body Vibration of Systemic Lupus Erythematosus Individuals: A Randomized Pilot Study

Carla F. Dionello [1,2], Patrícia L. Souza [1,2], Pedro V. Rosa [2], Andreza Santana [2], Renata Marchon [2,3], Danielle S. Morel [1,2], Eloá Moreira-Marconi [1,4], Eric F. F. Frederico [2,5], Danúbia C. Sá-Caputo [2,6], Ana Carolina Coelho-Oliveira [2,4], Marise Crivelli [7], Evandro Mendes Klumb [7], Redha Taiar [8,*], Pedro J. Marín [9] and Mario Bernardo-Filho [2]

1   Programa de Pós-Graduação em Ciências Médicas, Faculdade de Ciências Médicas, Universidade do Estado do Rio de Janeiro, Rio de Janeiro, RJ 20943-000, Brazil; carla_dionello@yahoo.com.br (C.F.D.); patricia.lopes.ppc@gmail.com (P.L.S.); daniellesoaresmorel@gmail.com (D.S.M.); eloamarconi@gmail.com (E.M.-M.)
2   Laboratório de Vibrações Mecânicas e Práticas Integrativas, Departamento de Biofísica e Biometria, Instituto de Biologia Roberto Alcântara Gomes and Policlínica Américo Piquet Carneiro, Universidade do Estado do Rio de Janeiro, Rio de Janeiro, RJ 20943-000, Brazil; pedrovitalrosa@gmail.com (P.V.R.); andrezatalk@gmail.com (A.S.); renatamarchon_fisio@hotmail.com (R.M.); ericfrederico@msn.com (E.F.F.F.); dradanubia@gmail.com (D.C.S.-C.); anacarol_coelho@hotmail.com (A.C.C.-O.); bernardofilhom@gmail.com (M.B.-F.)
3   Programa de Pós-graduação em Saúde, Medicina Laboratorial e Tecnologia Forense, Universidade do Estado do Rio de Janeiro, Rio de Janeiro, RJ 20943-000, Brazil
4   Programa de Pós-Graduação em Fisiopatologia Clínica e Experimental, Instituto de Biologia Roberto Alcântara Gomes, Universidade do Estado do Rio de Janeiro, Rio de Janeiro, RJ 20943-000, Brazil
5   Programa de Pós-Graduação em Biociências, Instituto de Biologia Roberto Alcântara Gomes, Universidade do Estado do Rio de Janeiro, Rio de Janeiro, RJ 20943-000, Brazil
6   Physiotherapy Department, Faculdade Bezerra de Araújo, Campo Grande, Rio de Janeiro, RJ 23052-090, Brazil
7   Hospital Universitário Pedro Ernesto (HUPE), Faculdade de Ciências Médicas, Universidade do Estado do Rio de Janeiro, Rio de Janeiro, RJ 20943-000, Brazil; marise.crivelli@terra.com (M.C.); klumb@uol.com.br (E.M.K.)
8   GRESPI, Physical and Rehabilitation Medicine Department, Sebastopol Hospital, University of Reims Champagne-Ardenne, 51100 Reims, France
9   CyMO Research Institute, 47012 Valladolid, Spain; pjmarin@checkyourmotion.com
*   Correspondence: redha.taiar@univ-reims.fr

**Abstract:** Background: Glucocorticoid induced osteoporosis (GIOP) is one of the most important causes of morbidity in lupus individuals. Whole body vibration exercises (WBVE) may be a safe alternative to prevent and amend muscular and bone damage, and decrease muscle related risk factors for falls. It is possible to evaluate neuromuscular responses to the WBVE through surface electromyography (sEMG). Objective: To analyze and compare the acute responses of the WBVE on sEMG of lower limbs of female systemic lupus erythematosus (SLE) individuals with chronic glucocorticoid use with and without bone impairments and non-lupus controls. Methods: All patients (non-lupus and with SLE) had a dual-energy X-ray absorptiometry (DXA) scan (body composition, bone composition right hip, lumbar segment and whole body). After DXA, they were divided into three groups: SLE with osteopenia (OPIA) (SLE OPIA), SLE without OP or OPIA (SLE) and non-lupus individuals as control (CG). Twenty-seven women were submitted to WBVE, on different frequencies with the same amplitude. The experiment was performed over two days, 48 h apart. The individuals stood at a half squat position on a vertical vibrating platform at different frequencies with the same amplitude on both days. Vastus lateralis (VL), gastrocnemius medialis (GM) and tibialis anterioris (TA) sEMG analyses were undertaken simultaneously while performing the exercises, in a randomized manner. Results: There were no differences between sarcopenia index among groups, despite the bone impairment of the SLE OPIA group. The greatest muscle activation occurred in the lower frequency applied for VL. A group x frequency difference was found only for GM ($p = 0.034$;

$\eta^2 = 0.272$). Conclusion: The results indicate that lupus individuals have similar neuromuscular activity to the WBVE as non-lupus controls. Moreover, this suggests that WBVE is a safe and viable physical exercise for lupus individuals with chronic glucocorticoid induced osteoporosis.

**Keywords:** lupus; osteoporosis; glucocorticoids; mechanical vibrations; exercises; muscle

## 1. Introduction

Systemic lupus erythematosus (SLE) is an inflammatory autoimmune disease that may affect any system of the body including the peripheral and central nervous system, skin, joints, blood, kidneys and serous membranes [1,2]. Chronic inflammation is the background for the various systemic manifestations. There is also prolonged endothelial damage, which impairs blood flow and causes cardiovascular events, which leads to increased mortality in patients who have had the disease long-term. The burden of the illness and its morbidity have being increasingly studied, since mortality has decreased in the past decades [3]. The treatment of individuals with lupus involves the suppression of disease activity [4], as well as the prevention of irreversible organ damage and related morbidities. For that purpose, several medications are employed but glucocorticoids (GCs) are still the cornerstone of treatment and multiple dosings of GCs over a lifespan is common [3]. Along with the inflammatory background, the chronic use of glucocorticoids directly impairs bone and muscle health through different mechanisms, including endocrine effects on calcium metabolism. GCs may cause hypocalcemia and affect parathormone (PTH) levels [5]. Prolonged glucocorticoid use reduces intestinal calcium absorption. Lower circulating levels of calcidiol may stimulate PTH secretion secondary to decreased calcium absorption or increased urinary calcium excretion, thus affecting bone remodeling [6]. There is also a reduction of osteoblast activity through decreases of their functional indices and useful time of action, especially on the pro-peptide of type I N-terminal (P1NP), pro-peptide of type I C-terminal (P1CP) and osteocalcin. There is decreased bone formation through inhibition of Dickkopf 1 protein (DKK-1) and sclerostin [5]. GC may also reduce osteoclastic activity, with increased production of receptor activator of nuclear factor kappa B ligand (RANK-L) and reduction of osteoprotegerin (OPG) [7,8]. Loss of bone mass might occur with GC use [9–11], and in contrast to the observations in postmenopausal osteoporosis (OP), fractures in glucocorticoid induced osteoporosis (GIOP) occur even at higher scores of bone mass [8]. With only 6 months use, there is already a reduction of bone mass in more than 25% of patients [7]. Vertebral and non-vertebral fractures are common, ranging from 30% to 50% in people who use GC for more than three months [10]. GC chronic use causes muscular catabolism and atrophy determined by a decrease in protein synthesis and proteolysis [12]. Thus, loss of muscular tissue, OP and fragility fractures should be prevented in all individuals who will initiate or who are already using these steroids [13].

Physical activities (PA) and exercises along with appropriate diet are reliable strategies to amend these morbidities [10]. In lupus, however, several questions related to optimal type of exercise to improve fracture risk parameters (muscle strength and bone density) are not yet answered [14]. Therapeutic exercise interventions to SLE patients appear to be safe, and not induce disease activity. Improvements on scores of fatigue, depression, and physical conditioning following exercise-based programs, such as stretching, resistance training, or aerobic exercise have been described [15]. It is known that not all exercises are equally effective in terms of improving musculoskeletal condition as well as consequences of GIOP, but no randomized controlled trials have shown a significant positive effect on bone mass and/or a reduction in fracture occurrence in individuals with SLE [11]. In fact, most publications involving exercise benefits in lupus, evaluated benefits on cardiovascular events and fatigue [15]. Since the disease may also cause pain, arthritis, damage to circulatory and nervous systems, and propensity to falls, along with a strong recommendation to avoid sun exposure, the safety when prescribing the exercise must also be considered [16].

Whole-body vibration exercises (WBVE) are a relatively recent type of physical exercise (PE) [15] that may provide specific benefits to mobility, functionality and muscular resistance [17–21]. It has already been demonstrated that SLE patients have diminished strength when compared to a healthy population. Several exercise modalities have been studied in lupus but not with the use of mechanical vibration [15]. Mechanical vibrations produced using a vibrating platform (VP) are transmitted to the body of the patient when in contact with the platform, generating the WBVE. In appropriate parameters, such as frequency ($f$), amplitude ($A$), peak-to-peak displacement ($D$), and peak acceleration (apeak), as proven in other diseases and fragile populations, it is important to demonstrate that WBVE is considered a secure and feasible form of physical activity for SLE [18]. Additionally, the chronic inflammation and possible drug side effects on muscles might affect the capacity to respond to this stimulation compared to controls [15].

The surface electromyography (sEMG) can be used to evaluate the effects of mechanical vibrations on muscles, so that it may be considered an appropriate method to analyze neuromuscular responses of muscle to WBVE and muscle fatigue [22–24].

The aim of this study was to compare acute effects during WBVE on lower limb muscles of lupus patients with and without decreased bone mass and non-lupus controls measured by sEMG. It was hypothesized that: (i) lupus and non-lupus individuals would show significant increases in lower limb activity during WBVE, (ii) those increases would be higher in the non-lupus group, and (iii) an association between the apeak and the sEMG activity may exist in all groups, but to a lesser extent in the lupus individuals with bone impairment.

## 2. Materials and Methods

A randomized, triple blinded, controlled, three-group parallel pilot study was designed. The 2010 Consolidated Standards of Reporting Trials (CONSORT) guidelines were used [25]. All participants were allocated 1:1 to either one of the four protocols of different orders of frequencies of mechanical vibrations upon enrollment (Figure 1).

This study was planned and designed with the international ethical guidelines for a clinical study. The research project was approved by the *Comissão Nacional de Ética em Pesquisa/Certificado de apresentação para Apreciação Ética* number 48116215.0.0000.5259. This clinical trial was registered in the Brazilian Clinical Trial Registry (ReBEC): RBR-2b4bzq (www.ensaiosclinicos.gov.br) according to international guidelines.

A sample size calculation (ß = 0.10) based on detecting a minimum of 20% of difference in sEMGrms of vastus lateralis (VL) (SD = 10) between the interventions, showed that at a 5% significance level would require a sample of 5 participants in groups with a power of 90% [26].

The lupus individuals were recruited from the Rheumatology Department of the *Universidade do Estado do Rio de Janeiro* (UERJ) in Rio de Janeiro, Brazil and were divided in two groups, SLE with OP or osteopenia (OPIA) (SLE OPIA) and SLE without OP or OPIA (SLE). They were all female outpatients undergoing their regular medical consultations on the lupus ambulatory, whom, after a brief interview, were invited to participate. Consenting patients were included if they were older than 40 years, diagnosed with SLE at least three years previously [27] and were regularly using glucocorticoids (prednisone or similar). The subjects must have been stable on lupus therapy, without recent increments on medications, with low index of disease activity and not having a disease flare according to clinical charts and the referring rheumatologist. All the medications for comorbidities were recorded, including diuretics and statins.

The non-lupus female participants (CG) were invited to the Rheumatology Department among the lupus patient acquaintances when the patients had their regular consultations at the hospital. They were interviewed and excluded if they had any inflammatory rheumatic disease or if they were regularly using GC.

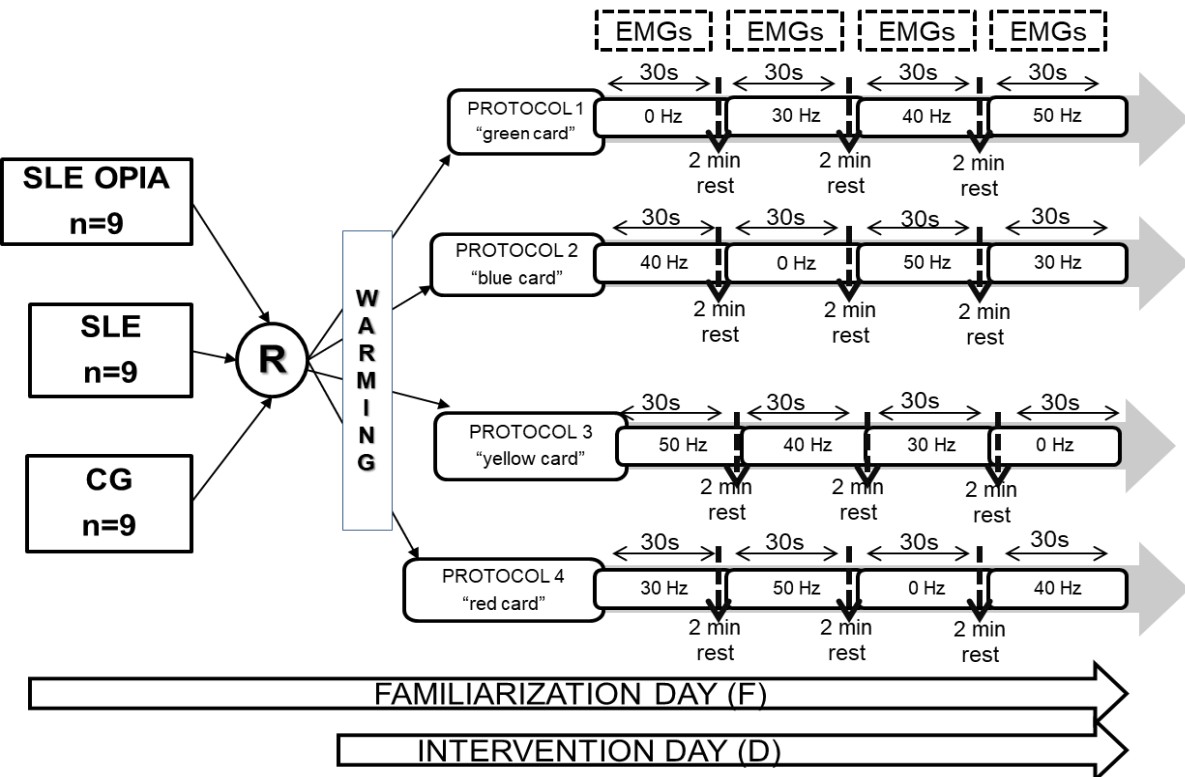

**Figure 1.** Whole body vibration exercise and surface electromyography sequences protocol. SLE OPIA—SLE individuals with OPIA, SLE—individuals without OP, CG—non-lupus individuals.

Patients were excluded if they smoked, had a history of alcohol abuse, history of low impact fractures, if they were using assistive walking devices, had hip or knee joint replacement surgery, and if they were pregnant. They were also excluded if their co-morbidities could potentially be affected by the PE, and if there was presence of neurological or psychiatric disease that caused fear to the movements on the VP.

The participants of all three groups (SLE OPIA, SLE and CG) who were eligible to participate and signed the informed consent were submitted to a dual X-ray absorptiometry (DXA). The scans assessed bone mineral density (BMD), T and Z scores of the right hip, lumbar spine (L1–L4), and total body. The T and or Z scores equal or less than −1.01 standard deviations (SD) in any of the regions of interest (ROI) were considered as compromise of bone mass as osteopenia is defined [28].

Subjects with SLE and loss of bone mass were classified to the SLE OPIA group. The lupus individuals without impairment of bone mass were allocated in the SLE group. The CG group was exclusively composed of non-lupus and non-bone mass reduction female individuals. If non-lupus participants had impairment of bone mass, they were excluded from the CG.

The regional and total lean/fat mass and derived indexes were obtained through analysis of whole body DXA [29–31]. For body mass index (BMI), the following ranges were used: normal, BMI = 18.5–24.9 kg/m$^2$; overweight, BMI = 25–29.9 kg/m$^2$; and obesity, BMI ≥ 30 kg/m$^2$, as recommended by the World Health Organization [32]. DXA body fat percentage (BF%) above 39% was considered to represent obesity since it has been demonstrated that in obese women, it ranges from 39 to 43%. The fat mass index (FMI) was also calculated and classified women as overweight if >9 kg/m$^2$ and obese if >13 kg/m$^2$ [33]. To evaluate the occurrence of sarcopenia, the skeletal mass index (SMI) below the specific mean of a healthy reference population (<5.5 kg/m$^2$ for women) was utilized [30].

All scans were performed by the same qualified technician who was unaware of the patients' diagnosis and on the same machine (GE Prodigy Advance H8610FE). The machine

was regularly calibrated, and a phantom spine was scanned daily to determine coefficients of variation [34].

After the division of groups, the participants came to the experimental Laboratory (*Laboratório de Vibrações Mecânicas e Práticas Integrativas*—LAVIMPI) to be submitted to the interventions. The entire protocol was performed in two days, the familiarization and intervention days; and all the groups were evaluated within a month. The randomization occurred on the familiarization day, in which every patient took a colored card kept in a brown envelope without looking inside. The WBVE sequential *f* order was according to the taken colored card (red, yellow, blue or green) (Figure 1). The physician who asked the participants to take the card was aware of the participant groups, but not the order of the frequency related to the card selected. The randomized patients were blinded on the f that they were being submitted, since the panel of the VP was kept covered during the whole working time. The physiotherapist that performed the intervention on VP knew the randomized frequency order of each randomized patient but was blinded to the group the patient belonged.

The study protocol is in accordance with the International Society of Musculoskeletal and Neuronal Interactions (ISMNI) recommendations for reporting WBVE studies [18]. All of the subjects were exposed to mechanical vibrations through a synchronous tri-planar VP Power Plate pro5 TM (Power Plate International LTD, The Netherlands). Patients were recommended to stand with knees flexed to 130° (half squat), with arms lose beside the body, and wearing their usual footwear, which was the same for both days of the evaluation (Figure 2). The physiotherapist remained near the patient for both safety purposes and to constantly check the knee angle of flexion during the WBVE.

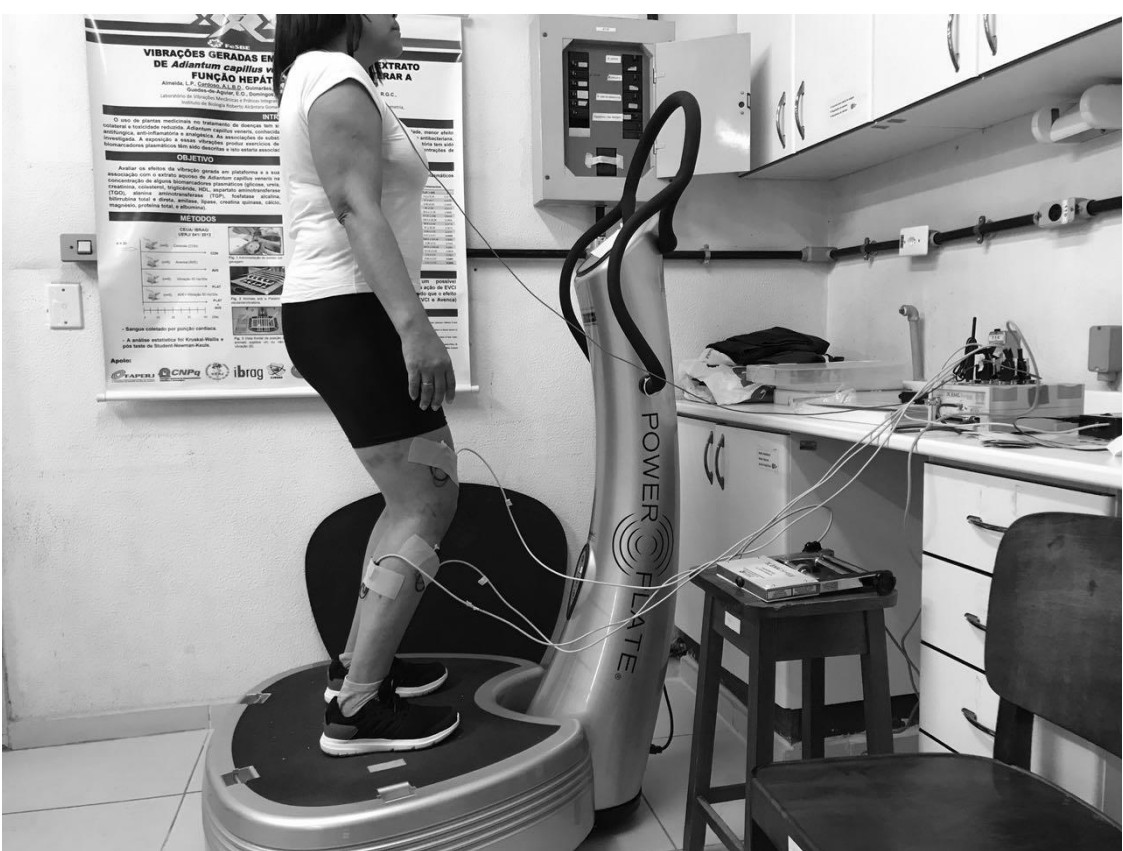

**Figure 2.** Positioning on the vibrating platform.

The full protocol was performed in the morning, after the participants had their usual breakfast, within the same month. The room, in which all the interventions were performed,

was kept at a constant temperature of 23 °C for the duration of the experiment. An interval of 48 h between familiarization and intervention days for every patient was applied. After a warm-up period of 5 min, which was performed with a cycloergometer, the patients were submitted to the same amplitude ("low") during the four different conditions of WBVE ($f$ = 0, 30, 40 and 50 Hz), in a randomized manner, as explained before (see Figure 1). The values of apeak were obtained by the equation apeak = $2 \times \pi^2 \times f^2 \times D$, or from the accelerometer, which was used in this study. The apeak during these four frequencies was respectively 0, 2.22, 3.06 and 4.40 g; measured using a three-axial accelerometer (Vibration Datalogger DT-178A, Ruby Electronics, Saratoga, NY, USA) (see Figure 1).

The sEMG to obtain muscle activity of vastus lateralis (VL), gastrocnemius medialis (GM), tibialis anterioris (TA) muscles was performed during the execution of WBVE according to Surface ElectroMyoGraphy for the Non-Invasive Assessment of Muscles (SENIAM) project recommendations (http://www.seniam.org).

Only the right lower limb was evaluated. The area of electrode placement was prepared before the protocol started. It was shaved and cleansed with 70% alcohol to reduce skin impedance. The double electrodes (Hal Indústria e Comércio Ltd.a, São Paulo/SP, Brazil) were placed according to the best location of each muscle surface. The electrodes were attached before the warm-up period. A ground electrode was placed over the C7 tuberosity vertebrae. The cables were fastened with kinesio tape (Kinesiology Tape Nitreat NKH-50BU, Nitto Denko Corporation, Osaka, Japan) to avoid motion artifacts. The surface electrodes were connected to a 16-bit AD converter (EMG System, São José dos Campos/SP, Brazil). Raw EMG signals were pre-amplified close to the electrodes (signal bandwidth of 10–500 Hz), sampled at 2000 Hz and stored on a laptop. sEMG data analysis was performed using specific computer software (EMG832WF, EMG System, São José dos Campos/SP, Brazil).

EMG raw data was equated by root mean square (EMGrms) in order to obtain the average amplitude of the EMG signal. Thirty seconds of signals were acquired, but the first and last 5 s were excluded, the central 20 s of signal acquisition were then analyzed [35].

*Statistical Analysis*

Data were analyzed using PASW/SPSS Statistics 24.0 (SPSS Inc., Chicago, IL, USA). The normality of the data was checked and subsequently confirmed with the Shapiro–Wilk test.

Comparisons of dependent variables (DXA and anthropometrics) between groups were evaluated with a one-way analysis of variance (ANOVA). Dependent variables (sEMG for each muscle) were evaluated with repeated measures analysis of variance (ANOVA). When a significant F-value was achieved, pairwise comparisons were performed using the Bonferroni post hoc procedure. The intra-class correlation coefficients (ICCs) were calculated for each sEMG dependent variable to determine test–retest reliability (between the last familiarization session and the test session). Statistical significance was set at $p \leq 0.05$. The ICCs were greater than 0.86, indicating that a high level of reproducibility in assessing the dependent variables was achieved. Effect sizes were measured by partial Eta square ($\eta^2$) to determine the magnitude of the effect independent of sample size. Values are expressed as mean $\pm$ SD in the text and tables, and as mean $\pm$ SEM in figures. Pearson correlation of body composition indexes was also performed.

## 3. Results

The participants' flow diagram of the study according to CONSORT 2010 statement may be observed in Figure 3. The experiment occurred in September 2017.

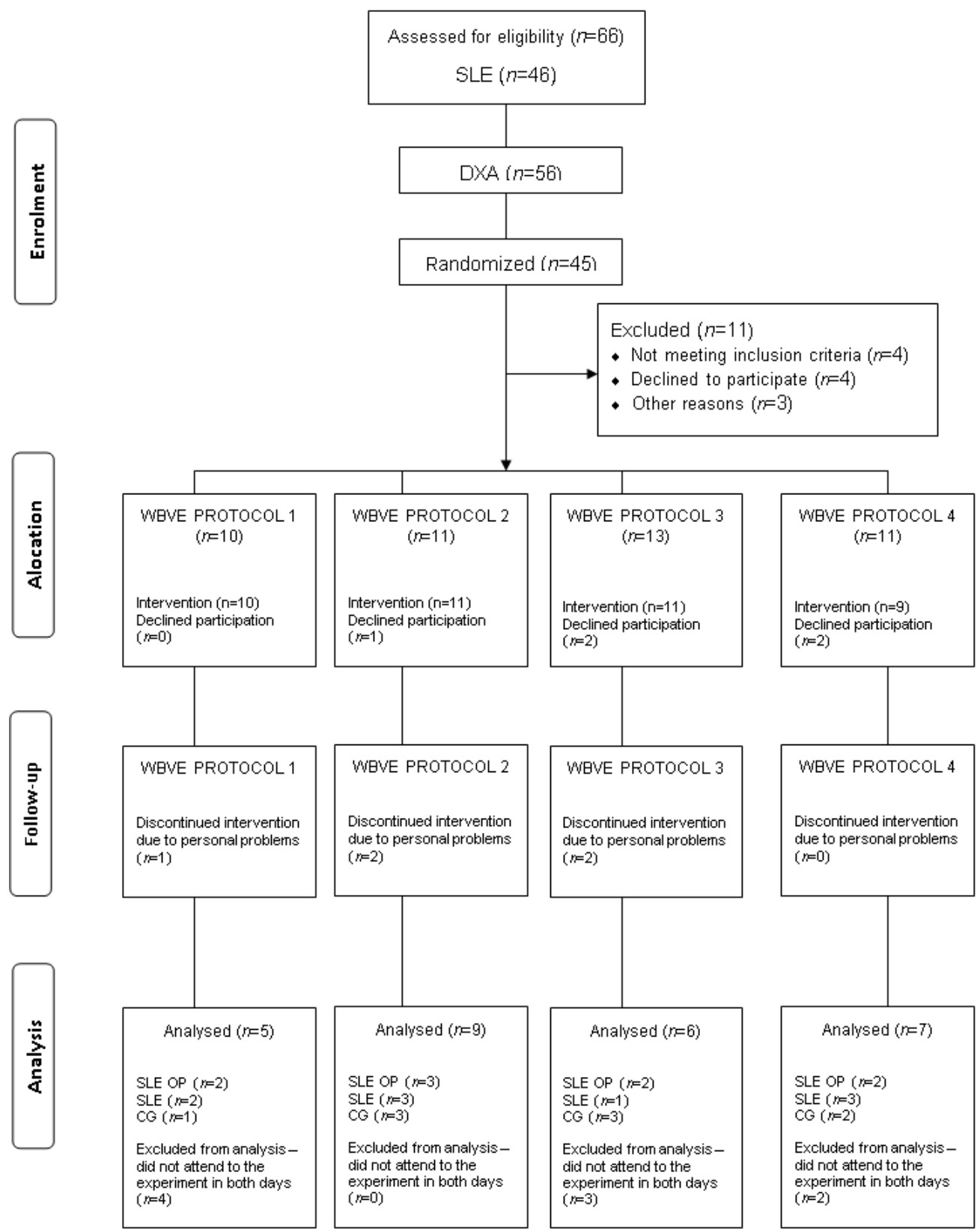

**Figure 3.** Flow diagram of the randomized clinical trial. DXA: dual X-ray absortiometry. SLE OPIA—SLE individuals with OPIA, SLE—individuals without OP or OPIA, CG—non-lupus individuals.

### 3.1. Baseline

Table 1 shows the baseline characteristics of the three groups. No significant difference was found ($p > 0.05$), including time of menopause and mean 6 months glucocorticoid dose equal or equivalent to prednisone, which might affect the body composition results and the sEMG evaluation.

**Table 1.** Baseline characteristics of the groups.

|  | SLEOPIA (n = 9) | | | SLE (n = 9) | | | CG (n = 9) | | |
|---|---|---|---|---|---|---|---|---|---|
| Age (y) | 51.3 | ± | 8.2 | 50.1 | ± | 6.95 | 50.63 | ± | 6.5 |
| Body mass (kg) | 69 | ± | 17.4 | 75.8 | ± | 16 | 73.4 | ± | 13.6 |
| Height (cm) | 150 | ± | 0.1 | 160 | ± | 0.1 | 160 | ± | 0.1 |
| Ethinicity | | | | | | | | | |
| caucasian (n/%) | 4 | / | 55 | 3 | / | 46 | 6 | / | 54 |
| non-caucasian (n/%) | 5 | / | 45 | 6 | / | 54 | 3 | / | 27 |
| Menopause (n/%) | 7 | / | 63 | 6 | / | 54 | 5 | / | 45 |
| Menopause time (y) | 8.2 | ± | 8.3 | 3.9 | ± | 5.1 | 2.9 | ± | 4.8 |
| Diuretics (n/%) | 4 | / | 36 | 3 | / | 27 | 2 | / | 18 |
| Statins (n/%) | 2 | / | 18 | 2 | / | 18 | 1 | / | 9 |
| Lupus time of diagnosis (y) | 18 | ± | 9.5 | 19.1 | ± | 11.7 | | n.a | |
| Lupus treatment: | | | | | | | | | |
| prednisone * (mg; Δ daily dose) | 4.0 | ± | 2.6 | 4.8 | ± | 2.2 | | n.a | |
| prednisone * (mg; Δ 6 m cumulative dose) | 679.7 | ± | 444.2 | 799.7 | ± | 365.3 | | n.a | |
| prednisone * time of use (y) | 18.6 | ± | 8.9 | 18.3 | ± | 9.1 | | n.a | |
| hydroxicloroquine (n/%) | 8 | / | 72 | 7 | / | 63 | | n.a | |
| immunosuppressants (n/%) | 9 | / | 100 | 8 | / | 72 | | n.a | |

SLE OPIA—SLE individuals with OPIA, SLE—individuals without OP or OPIA, CG—non-lupus individuals; Data are presented in mean and and Standard error of mean (SEM). n.a—not applicable; * or equivalent GC; $p > 0.05$.

### 3.2. Body Composition

Table 2 demonstrates significant differences in bone mass parameters for the three groups, where SLE OPIA has the lower T and Z scores in every ROI, especially on the lumbar segment.

**Table 2.** DXA bone, lean and fat tissues results according to each group.

| Groups | T Score L1–L4 | Z Score L1–L4 | T Score Neck | Z Score Neck | T Score Total Body | Z Score Total Body |
|---|---|---|---|---|---|---|
| **SLE OPIA** | −2.17 ± 0.71 | −1.88 ± 1.20 | −1.50 ± 0.83 | −0.93 ± 0.61 | −1.16 ± 0.89 | −0.90 ± 0.69 |
| **SLE** | 0.10 ± 0.73 | −0.03 ± 0.31 | −0.08 ± 1.11 | 0.24 ± 0.60 | 0.72 ± 0.87 | 0.39 ± 0.59 |
| **CG** | −0.09 ± 1.37 | −0.03 ± 1.16 | −0.24 ± 1.20 | 0.26 ± 1.03 | 0.29 ± 1.14 | 0.19 ± 0.90 |
| ***p* value** | < 0.0001 | 0.0006 | 0.0179 | 0.0038 | 0.0018 | 0.0035 |

ASMM = appendicular skeletal muscle mass (arms + legs); BF% = body fat percentage; BMI = body mass index (weight/height$^2$); FMI = fat mass index (total body fat/height$^2$); SMI = skeletal mass index (ASMM/height$^2$), SLE OPIA—SLE individuals with OPIA, SLE—individuals without OP or OPIA, CG—non-lupus individuals. Statistical test: ANOVA, $p < 0.05$. Data are presented in mean ± SD.

As demonstrated in Figure 4, it is possible to observe that of the three ROIs included in the analysis, the group SLE OPIA showed a lower BMD in the lumbar segment compared to the other two groups, as expected.

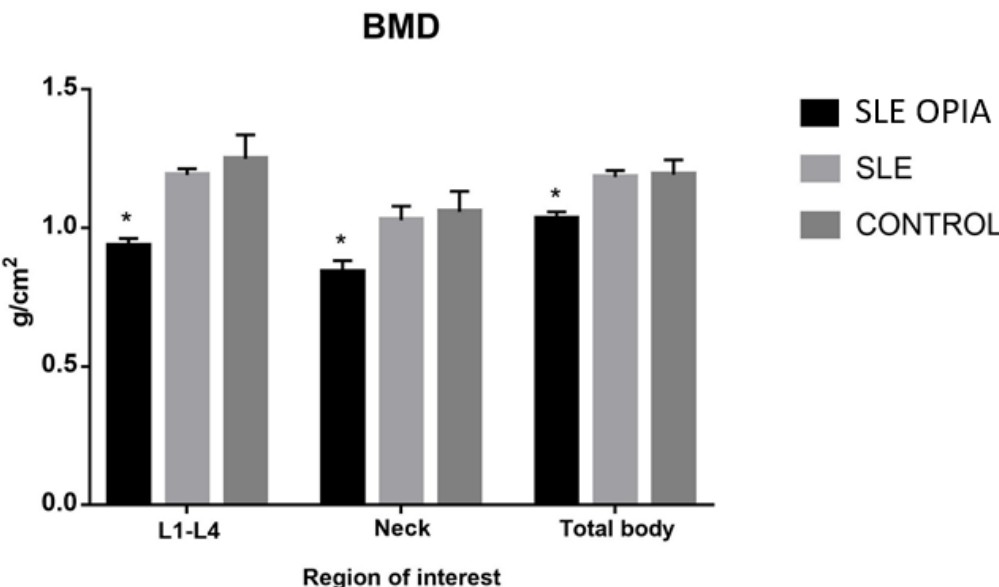

**Figure 4.** Bone mineral density comparison among the groups. SLE OPIA—SLE individuals with OPIA, SLE—individuals without OP or OPIA, CG—non-lupus individuals. BMD—bone mineral density. Statistical test: ANOVA. * $p < 0.05$.

The mean lumbar BMD was $0.925 \pm 0.075$ g/cm$^2$ for SLE OPIA, $1.226 \pm 0.115$ g/cm$^2$ for SLE and $1.184 \pm 0.172$ g/cm$^2$ for CG ($p < 0.001$). The mean femur neck BMD was $0.844 \pm 0.098$ g/cm$^2$ for SLE OPIA, $1.020 \pm 0.129$ g/cm$^2$ for SLE and $1.006 \pm 0.165$ g/cm$^2$ for CG ($p = 0.0332$). The mean total body BMD was $1.045 \pm 0.061$ g/cm$^2$ for SLE OPIA, $1.198 \pm 0.064$ g/cm$^2$ for SLE and $1.148 \pm 0.091$ g/cm$^2$ for CG ($p = 0.0018$). In Table 2, the lean and fat tissue composition of the three groups can be observed, with several indexes used to analyze obesity and sarcopenia [30]. No significant differences in the calculations for each group were found, all the participants being similar in fat and lean tissue composition, albeit the low bone mass of SLE OPIA group.

The BMI of the three groups demonstrated no obese subjects, only overweight, for the three groups (<30 kg/m$^2$). That finding was different from BF% and FMI. BF% was above 43% for all patients; according to this index, all the participants were obese, but the mean FMI suggested overweight for SLE OPIA (<13 kg/m$^2$) and class I obesity for SLE and CG (>13 kg/m$^2$). The BMI and the SMI were correlated as expected in SLE OPIA ($r = 0.892$; $p = 0.003$) and CG ($r = 0.939$; $p = 0.001$) groups, but not for SLE ($r = 0.595$; $p = 0.09$). None of the three groups demonstrated sarcopenia ($\Delta$SMI > 5.5 kg/m$^2$), but the BMD L1–L4 was significantly negatively correlated to SMI for SLE OPIA ($r = -0.759$; $p = 0.029$).

### 3.3. Muscular Activation of Lower Limbs

Figure 5 shows the percentage of increase in muscle activation (EMGrms) compared to no vibration (0 Hz) evaluated by sEMG of GM, VL and TA muscles.

The GM had a significant increase in activation on 30 Hz for the CG ($p = 0.034$; $\eta^2 = 0.272$), but differences for the other groups or other frequencies were not observed.

The VL had a significant increase in activation during WBVE executed in 30 Hz for all three groups ($p = 0.02$; $\eta^2 = 0.388$) but not the specific interaction "group versus frequency" ($p$ = n.s.). Regarding TA, no significant increases in muscular activation comparing groups ($p = 0.271$; $\eta^2 = 0.151$) or f ($p = 0.062$; $\eta^2 = 0.294$) were observed during WBVE.

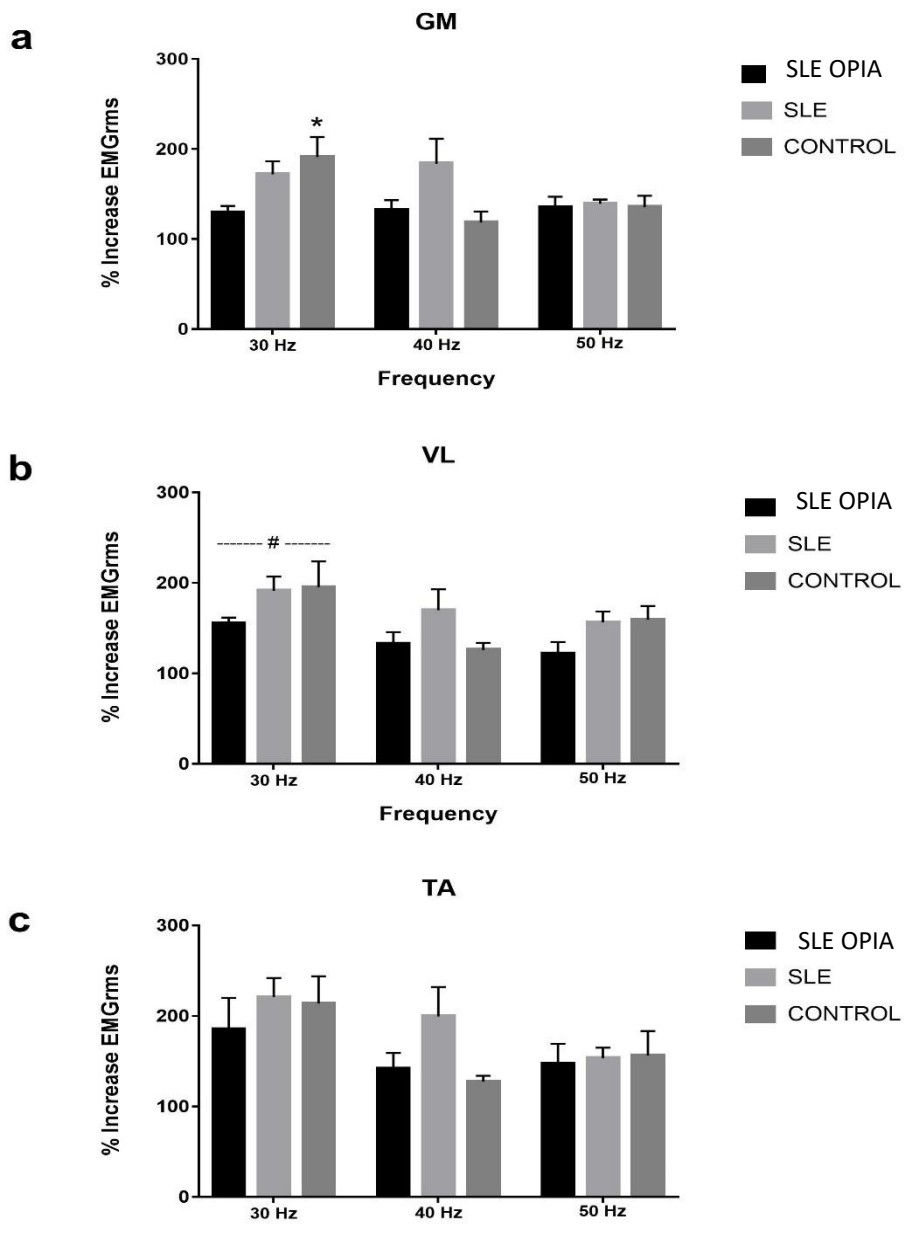

**Figure 5.** sEMG analysis of muscles gastrocnemius medialis, vastus lateralis and tibialis anterioris. SLE OPIA—SLE individuals with OPIA, SLE—individuals without OP or OPIA, CG—non-lupus individuals. GM—gastrocnemius medialis, VL—Vastus lateralis, TA—tibialis anterioris. Statistical test: ANOVA. *,# $p < 0.05$. (**a**): % increase in EMGrms of GM, (**b**): % increase in EMGrms of VL, (**c**): % increase in EMGrms of TA.

## 4. Discussion

Following the results of muscle activity secondary to WBVE measured by sEMG in SLE individuals, longer duration WBVE protocols may be planned, and a prolonged intervention might help amend muscle and bone loss in this diseased population.

Caution should be taken when determining overweight or obesity in inflammatory rheumatic diseases and lupus individuals with chronic GC using only BMI since it was demonstrated that there was a discrepancy between BMI, BF% and FMI [33,36]. These characteristics may misestimate morbidity in this population.

Mak 2019 [15] pointed out that despite the cardiovascular and psychosocial benefits conferred by physical activity, SLE individuals do not exercise adequately in general.

In the current study, considering the characteristics of the biological responses of the WBVE [17–20], this modality of exercise might be of benefit in the management of SLE individuals.

Lupus individuals had acute increases in activation of VL, GM and TA during WBVE measured by sEMG. In respect of the hypotheses, all three groups demonstrated similar increased activation of VL in the lowest frequency implemented in this investigation, 30 Hz; GM demonstrated the best activation in 30 Hz, especially for controls.

To our knowledge, this is the first study that evaluated WBVE effects on lupus individuals. The importance of studying different categories of PE relies on the fact that recommendations of exercises to amend decreased mobility and bone loss should be tailored to each group of diseases' needs and peculiarities [31]; SLE patients cannot be sun exposed and may have movement impairments secondary to arthritis, circulatory and neurologic sequelae [35]. Although there are several effective medications that are utilized to treat GIOP [10,37,38], these agents have no effect on other key fracture risk factors, such as muscle weakness, diminished functionality and altered body composition.

The interaction of mechanical vibration with the structures of the body would induce the process of muscle activation and bone formation [39]. Most studies on WBVE for OP treatment were performed with post-menopause, elderly, institutionalized patients [40–42] and several of the obtained outcomes demonstrated an increase in BMD, better functionality and enhanced muscle strength and power [43–46]. A recent systematic review on children with Duchenne muscular dystrophy with chronic GC use demonstrated improvements in bone mass only with WBVE and risedronate, a commonly used drug for the treatment of OP [47]. As muscle stimulation and bone remodeling are directly related, it is plausible to evaluate two groups of the same disease that differ on bone mass. The occurrence of osteoporosis could affect muscle functioning so that mechanical vibration would not demonstrate the same effects. A single different parameter on intervention might show this distinction, especially frequency, which has the most impact on apeak results [48].

The mechanisms by which the mechanical vibrations act on muscle would be associated to the stretch reflex [49]. Increases on sEMGrms for lower limb muscles of older populations when compared to young individuals were prominent [50,51]. The authors relate the lower strength of old adults to the higher activation while standing on the functioning VP, as an attempt of adaptation, which may further improve stability and muscle power [26].

Although chronic GC use is associated with muscle deterioration [52], the individuals of the current investigation did not demonstrate differences in lean mass composition, even the SLE OPIA group having bone waste. This may be responsible for the similar muscular activation among the three groups, except for GM.

Studies have demonstrated an increase in VL activation with 30 Hz [53,54], similar to this current investigation, but most of these studies were performed in healthy subjects. A study that compared aerobic exercise to WBVE in different frequencies for obese patients showed that the activation of VL on 30 Hz improved results on the stair-climbing test [55].

The demonstration of neuromuscular activity of lower limbs in populations with a medical condition during WBVE is useful and may provide information independently of the pathophysiological mechanism of the disease [23]. Even paretic or spastic lower limbs may be activated during these exercises, but especially in higher apeak values [24,56]. The findings of these publications were contrary to the present study, since higher f demonstrated lower VL sEMGrms. Another study in Friedreich's ataxia showed a better result in lower *f*, analogous to this investigation [57]. The molecular effects of GC on muscle that may alter responses to impact and mechanical vibrations are still to be answered, but the sEMG responses seem to be like non-GC users according to these results.

*Study Limitations*

As a potential limitation, this study focused only on acute effects. Still to be determined are the chronic and cumulative effects of WBVE in lupus individuals, regarding bone,

muscles, quality of life and functional outcomes. Only three different *f* and a small sample were evaluated; perhaps with another *f* and a larger number of patients, other findings might possibly be achieved. Another limitation is that only a vertical VP was used; different results might have been observed with an alternating platform.

## 5. Conclusions

It was demonstrated that lupus individuals might benefit from WBVE in similar protocols of healthy women of the same age, as the acute increases on lower limb muscle activity during WBV are alike, especially for VL. These findings may be encouraging for the practice. There were no adverse events during the intervention, or thereafter. There was no increase in pain or fatigue. There were neither reported headaches nor vascular symptoms such as spontaneous ecchymosis. The patients finished the two-day protocol. After the results of this study, it is possible to suggest starting a protocol for SLE patients of 6 or 12 weeks (low amplitude; 30 Hz or lower f) with progressive increases in workload, using a vertical VP.

If the practice of PE is considered as complimentary to a medical prescription with a progression plan, it will be possible to obtain the desired outcome as well as the patient wellbeing and compliance. This is highly desired in populations of higher risk of obesity, muscle weakness and fractures in young age, like lupus individuals, and this may be possible in optimized conditions, with WBVE, that is performed indoors.

**Author Contributions:** Conceptualization, C.F.D., P.L.S., A.S., D.C.S.-C. and M.B.-F.; methodology, C.F.D., P.L.S.; software, C.F.D.; validation, C.F.D., P.L.S., D.C.S-C. and M.B.-F.; formal analysis, C.F.D., R.M.; investigation, C.F.D., P.L.S. and A.S.; resources, C.F.D. and P.L.S.; data curation, C.F.D., R.M.; writing—original draft preparation, C.F.D.; writing—review and editing, E.M.-M., D.S.M., E.F.F.F., P.V.R., E.M.K., M.C., R.T., A.C.C.-O., D.C.S.-C., M.B.-F. and P.J.M.; visualization, E.M.-M., D.S.M., E.F.F.F., P.V.R., E.M.K., M.C., R.T., A.C.C.-O., D.C.S.-C., M.B.-F. and P.J.M.; supervision, D.C.S.-C.; project administration, M.B.-F.; funding acquisition, M.B.-F. All authors have read and agreed to the published version of the manuscript.

**Funding:** This research was funded in Brazil: Fundação Carlos Chagas Filho de Amparo à Pesquisa do Estado do Rio de Janeiro (FAPERJ), Conselho Nacional de Desenvolvimento Científico e Tecnológico (CNPq) and Coordenação de Aperfeiçoamento de Pessoal de Nível Superior—Brazil (CAPES)—Finance Code 001.

**Institutional Review Board Statement:** The study was conducted according to the guidelines of the Declaration of Helsinki, and approved by the Ethics Committee *Comitê de Ética em Pesquisa do Hospital Universitario Pedro Ernesto–UERJ* (protocol code 1.274.552 on 27 July 2015).

**Informed Consent Statement:** Informed consent was obtained from all subjects involved in the study.

**Data Availability Statement:** Not applicable.

**Conflicts of Interest:** The authors declare no conflict of interest.

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
