# Peer review of "Acute Neuromuscular Responses to Whole-Body Vibration of Systemic Lupus Erythematosus Individuals: A Randomized Pilot Study"

_applsci, doi:10.3390/app11010138_

Round 1
Reviewer 1 Report
In this randomized clinical trial, authors investigated effects of whole body vibration exercises on neuromuscular responses in patients with systemic lupus erythematosus. Same amplitude with different frequencies was used, and neuromuscular responses were collected form vastus lateralis, gastrocnemius medialis and tibialis anterior using surface electromyography.
As it appears that effects of WBVE in lupus patients are not yet described in this manner in literature, this study provides some information (according to relatively small sample size) about benefits of this type of exercise. However, there are several important concerns and key questions to be answered before potential publishing, and overall improvement of the study is needed.
Study could be written in better English language. There are many grammatical errors and confusingly written sentences from which clear message is not easily understood. Thorough checkup and improvement is advised, preferably from a professional.
Whole body vibration exercise is already described in literature as one of the recommended physical activities for osteoporosis and other conditions. Here, it is confirmed that lupus patients respond to this type of exercise as well, not differently from healthy individuals. Is there any evidence that authors could point that lupus patients would not respond to WBVE, or that results would be significantly different in observed groups (as authors hypothesized)? Elaborate.
Explain in more detail why it is important to this study to compare SLE groups with and without impaired bone health, as authors only assessed acute muscular responses on different frequencies?
In SLE patients, are there other mechanisms that could have deleterious effect on bone health (inflammation etc.), other than GC treatment? It is not addressed in the paper.
In abstract, authors state that “WBVE is a safe and viable physical exercise for lupus individuals with chronic glucocorticoid”. Why it wouldn’t be safe, and how exactly findings from this study provide sufficient evidence for this statement? The same is with conclusions in discussion section (lines 336-337, and study is made on small sample size). Elaborate in detail, or change more appropriately.
Line 73 – explain GC effect on PTH in more detail
Line 79-85 – authors only explained GC effect on bone mass, and not on trabecular microarhitecture. It is important to address this topic, as fragility fractures are connected to bone quality as well.
Line 86-88 – provided reference is not addressing enough physical exercise, but only fractures in SLE patients. Explain this part in more detail, and provide references and explanations from other studies that investigated physical activity in SLE.
Line 94-96 – What other exercise types would have similar effect as WBVE, according to mentioned risk factors? Why would WBVE be potentially better than those, as main cause of bone health improvement would be muscle activation as well?
Line 100-103 – explain mentioned WBVE parameters in more detail
Is there another WBVE mechanism that could have beneficial bone health effect, in addition to muscle activation?
Line 111 – authors did not elaborate enough hypotheses III in results, or discussion section
In author’s opinion, could results from this study be associated with any patient that is on GC treatment?
Lines 136-139 – Explain in more detail the process of CG selection
Authors chose that group with bone impairment would be called “SLE-osteoporosis” group. I recommend changing this phrase, as osteoporosis is defined as T-score under -2.5 SD, and not ‑1.0 SD, like authors defined it. This could be confusing to the readers, as result between -2.5 SD and -1.0 SD is defined as osteopenia.
In the methods section, it would be interesting to add relevant picture of the WBVE procedure.
Line 214-216 – as this study is made on small sample size, it could be more appropriate to use non-parametric statistical tests
Is this muscle activity enough to improve bone health? Elaborate better in discussion section.
Figure 3 is not needed, as it proposes obvious results. It can be incorporated into Table 2
Could there be any connection between body composition indexes and muscular activation? Did authors made statistical analysis of these parameters, as it could provide interesting results?
Line 239 – this sentence should be in methods section
Line 306-308 – elaborate better the importance of VL activation on 30Hz, as it proposes one of more important findings in this study. Furthermore, more detailed comparison with other studies is needed.
Figure 4 – explained abbreviations and used statistical tests are missing from the figure legend
Table 2 – used statistical test is lacking in legend under the table
All incorporated figures in this manuscript are in very poor quality. I recommend re-upload of all figures.
Reviewer 2 Report
Congratulations to the authors for this interesting article. The objective is well covered and I think it may be of interest to clinicians. Still, I think that some changes should be considered for a better presentation.
First, I suggest that the authors treat this study as a pilot study, given their small sample size.
Second, I suggest using a tempered manuscript of the journal to better fit that format, and, for example, putting the tables and figures in the correct place in the text and not at the end.
Third, both in introduction and discussion sections, I think too many paragraphs are abused. Many can go together and not have 9 paragraphs in the introduction.
Fourth, the language must be checked. For example, the oxford comma is not used.
Fifth, a conclusion section is necessary.
Reviewer 3 Report
The introduction could be made more compact. The article still needs to be checked by a native speaker.Author Response
Please see the attachment.

Round 2
Reviewer 1 Report
Authors have been responsive, and manuscript is overall improved. I find all questions adequately answered and incorporated into manuscript.
Only problem was finding all exact changes, as authors didn't specify in what exact lines of the text changes were made.
I have no further questions, and now I find this manuscript adequate for publication.